# Prevalence of infantile wheezing and eczema in a metropolitan city in Japan: A complete census survey

**Masaki Futamura** [1]*, **Yoshimichi Hiramitsu**[2], **Naomi Kamioka**[3], **Chikae Yamaguchi**[4], **Harue Umemura**[5], **Rieko Nakanishi**[6], **Shiro Sugiura**[7], **Yasuto Kondo**[8], **Komei Ito** [7]

1 Department of Pediatrics, National Hospital Organization Nagoya Medical Center, Aichi, Japan, 2 Nagoya City Public Health Research Institute, Aichi, Japan, 3 Department of Pediatrics, Nagoya City University West Medical Center, Aichi, Japan, 4 Department of Community Health Nursing, Nagoya City University Graduate School of Nursing, Aichi, Japan, 5 School of Nutritional Sciences, Nagoya University of Arts and Sciences, Aichi, Japan, 6 Allergy Support Network, Nagoya, Japan, 7 Aichi Children's Health and Medical Center, Aichi, Japan, 8 Department of Pediatrics, Fujita Health University Bantane Hospital, Aichi, Japan

* masakifutamura@gmail.com

## Abstract

Infantile wheezing and eczema are associated with the subsequent onset of asthma and other atopic diseases. However, there are no large population-based surveys on infantile allergic symptoms in Japan. The objective of the study was to determine the prevalence of wheezing and asthma in infants in Nagoya, Japan. This population-based cross-sectional study was performed in the metropolitan city of Nagoya, Japan. We surveyed parents to ascertain the prevalence of wheezing and eczema in infants who attended group health checkups at 3, 18, and 36 months of age. Their parents completed modified questionnaires from the International Study of Asthma and Allergies in Childhood. More than 90% of the approximately 40,000 children in each study group living in the target area were included in the survey. The prevalence of wheezing was 8%, 17%, and 13% at 3, 18, and 36 months, respectively, and was characterized by birth season. The prevalence of eczema was 24%, 30%, and 31%, at 3, 18, and 36 months, respectively. Participants born in autumn and winter had a higher incidence of eczema in each age group. Three-quarters of the children had a parental history of allergic conditions. Parental allergic diseases and male gender are risk factors for wheezing and eczema in children. This survey had a high response rate and covered almost the entire population of the target age groups in a large city. We believe that the results of this study, therefore, provide a much higher level of confidence regarding the prevalence of allergies in infants in Japan than that in previous studies with limited cohorts.

## Introduction

Many children suffer from allergic diseases that affect their quality of life. In Japan, the national prevalence of asthma and atopic dermatitis has been found to be approximately 10% [1]. Adult patients who suffer from allergic diseases often experience an economic burden [2, 3]. Due to

**Data Availability Statement:** Data cannot be shared publicly according to the rule of the local government who provided us the data. Data are available from the Division of Environment Disaster

and Health, Environmental Bureau of Nagoya (www.city.nagoya.jp/en/), for researchers who meet the criteria for access to confidential data.

**Funding:** The authors received no specific funding for this work.

**Competing interests:** The authors have declared that no competing interests exist.

the high prevalence of allergies in Japan and the resulting social problem, a law outlining measures against allergic diseases came into force in 2014 [4].

Allergic diseases are chronic conditions that can be well controlled but are difficult to cure completely. Several studies on allergy prevention have been conducted worldwide. However, only few trials have proven the effectiveness of preventive strategies [5, 6].

The occurrence of wheezing and eczema in infancy is strongly associated with the subsequent onset of asthma and other allergic diseases [7, 8]. Large and carefully constructed observational surveys are needed to identify risk factors for the onset of atopic disease. To date, in Japan, there have been no large-scale surveys of allergies in the entire population of a region.

In Japan, an infant checkup program has been established by law to improve public health. Many local governments organize group-based health checkups for infants, and most parents agree to participate. The government of Nagoya, a metropolitan city in central Japan, conducts group health checks for infants at the ages of 3, 18, and 36 months, and this includes the collection of information on any allergy symptoms.

This study aimed to determine the prevalence of allergic symptoms in Japanese infants in a large metropolitan area using the data from a parent-answered questionnaire.

## Methods

### Study population

This study involved the analysis of data from a cross-sectional population survey of Japanese children. We collected data from the parents of children attending group health checks in Nagoya City between April 2016 and March 2018. Nagoya is home to 2.3 million citizens, including 280,000 children. It is Japan's fourth-largest city and is located in central Japan. Group health checks are regularly conducted in 16 residential districts of Nagoya for children at 3, 18, and 36 months old, who attend these checkups either in the month they reach the age target or in the following month.

### Questionnaire

The 12-month prevalence of wheezing and eczema was investigated in each age group using a modified questionnaire from the International Study of Asthma and Allergies in Childhood (ISAAC) [9]. We defined wheezing and eczema when a parent answered yes to the questions, "Has your child had wheezing or whistling in the last 12 months?" and "Has your child had itchy eczema with repeated appearance and disappearance in the last 12 months?", respectively.

In the 3-month-old group, the lifetime allergy history from birth was investigated. The questionnaires included questions on sex, the month of birth, existence of older siblings, and parental history of allergic diseases. Parental history of allergic diseases was defined as self-reporting of doctor-diagnosed asthma, atopic dermatitis, food allergies, anaphylaxis, and allergic rhino-conjunctivitis.

### Data collection

Questionnaires were mailed to parents in advance of each health checkup. The parents voluntarily answered the questionnaires, which were collected by the government health officials when the children participated in the checkups, and we were provided with the data from children in each age group. Since participation in the checkup was permitted in the appropriate and subsequent months, data from two consecutive years were collected to minimize variations in the number of participants born in a specific month.

## Statistical analysis

Data were analyzed using SPSS version 27 (IBM, Armonk, NY, USA), with a p-value of <0.05 defined as statistically significant. We categorized birth seasons as follows: spring from March to May, summer from June to August, autumn from September to November, and winter from December to February. The prevalence of allergic symptoms was calculated in each age group. Logistic regression was used to estimate crude odds ratios (ORs) and their 95% confidence intervals (CIs) for the associations between allergic symptoms and sex, birth season, birth order, and parental allergic diseases. Multivariable models were constructed for each outcome. All confounders, except parental allergic diseases, were considered for inclusion.

## Ethics

This study was approved by the Institutional Review Board of Aichi Children's Health and Medical Center (2019075). Parents of participants provided written consents on the questionnaire for using the data for epidemiological studies. We were supplied with the data by the government without any identifying information for individual participants. Anonymous data were managed by the government of Nagoya.

## Results

During the two years study period, group checkups were held for 40,242 infants aged 3 months, 40,175 infants aged 18 months, and 39,136 infants aged 36 months. Of these, questionnaires were independently collected from the parents of 39,202 (97%) infants at 3 months, 38,609 (96%) at 18 months, and 37,319 (95%) at 3 years, with completed questionnaires accounting for 90% of the infant population living in the target area (Fig 1).

Participants' characteristics are presented in Table 1. Half of the participants were women, 52% of them were first-born babies, and participants were born equally across each season. Three-quarters of the participants had a parent with a history of allergic disease. Allergic rhino-conjunctivitis was the most dominant allergic disease in parents, with more than 60% of parents affected. In infants, the prevalence rates of wheezing were 8%, 17%, and 13%, those of

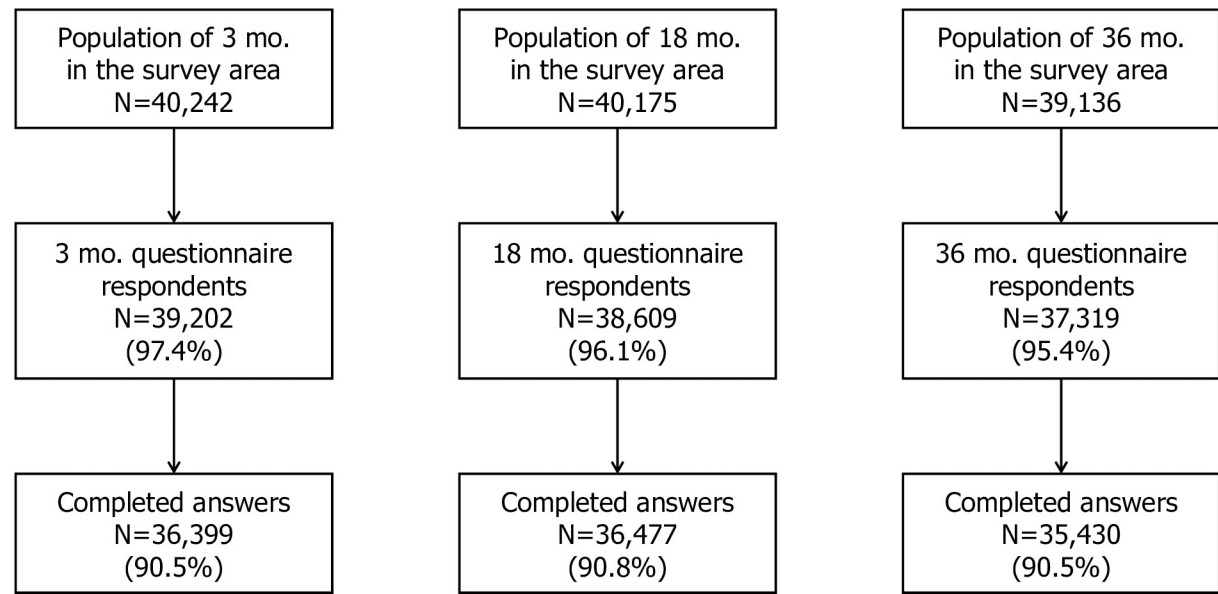

**Fig 1. Population in the survey area and response rates of questionnaire in three age groups.** (mo = months old).

**Table 1. Characteristics of the study population.**

| Characteristics | | 3 month group | (%) | 18 month group | (%) | 36 month group | (%) |
|---|---|---|---|---|---|---|---|
| Total | | 36,333 | | 36,391 | | 35,387 | |
| Gender | | | | | | | |
| | Female | 17,652 | 49 | 17,772 | 49 | 17,122 | 48 |
| | Male | 18,681 | 51 | 18,619 | 51 | 18,265 | 52 |
| Birth season | | | | | | | |
| | Spring (Mar. to May) | 9,014 | 25 | 9,104 | 25 | 8,447 | 24 |
| | Summer (Jun. to Aug.) | 9,286 | 26 | 9,192 | 25 | 9,236 | 26 |
| | Autumn (Sep. to Nov.) | 9,125 | 25 | 9,063 | 25 | 9,216 | 26 |
| | Winter (Dec. to Feb.) | 8,908 | 25 | 9,032 | 25 | 8,488 | 24 |
| Birth order | | | | | | | |
| | First | 18,731 | 52 | 18,941 | 52 | 18,370 | 52 |
| | Second and after | 17,602 | 48 | 17,450 | 48 | 17,017 | 48 |
| Parental allergic disease | | | | | | | |
| | Any | 26,806 | 74 | 25,889 | 71 | 24,987 | 71 |
| | Allergic rhinoconjunctivitis | 23,281 | 64 | 22,333 | 61 | 21,604 | 61 |
| | Atopic dermatitis | 9,931 | 27 | 9,348 | 26 | 8,808 | 25 |
| | Bronchial asthma | 5,668 | 16 | 5,215 | 14 | 5,056 | 14 |
| | Food allergy | 3,918 | 11 | 3,495 | 10 | 2,944 | 8 |
| | No | 9,527 | 26 | 10,502 | 29 | 10,400 | 29 |
| Current wheeze | | | | | | | |
| | Yes | 2,929 | 8 | 6,131 | 17 | 4,742 | 13 |
| | No | 33,404 | 92 | 30,260 | 83 | 30,645 | 87 |
| Current eczema | | | | | | | |
| | Yes | 8,562 | 24 | 10,785 | 30 | 10,909 | 31 |
| | No | 27,771 | 76 | 25,606 | 70 | 24,478 | 69 |
| Current wheeze and eczema | | | | | | | |
| | Yes | 860 | 2 | 2,353 | 7 | 2,145 | 6 |
| | No | 35,473 | 98 | 34,038 | 94 | 33,242 | 94 |

eczema were 24%, 30%, and 31%, and those of both symptoms were 2%, 7%, and 6% at 3, 18, and 36 months, respectively.

Analysis of seasonal factors showed different tendencies in each age group. More wheezing was recorded in 3-month-old participants born in summer and autumn, in 18-month-old participants born in spring, and 36-month-old participants born in autumn. Winter participants born in autumn and winter had more eczema in all age groups, but the difference was smaller in the older group. The tendency in eczema was also observed in both symptoms (Fig 2).

The prevalence of wheezing was significantly higher in male participants in all age groups (Table 2). Children with older siblings also had a significantly higher prevalence of wheezing than first-born babies at three months but not in the older age groups.

Like wheezing, the prevalence of eczema was also significantly higher in male participants in all age groups, while first-born children had a significantly higher prevalence of eczema in all age groups (Table 3). The significant difference in the first-born children disappeared for both symptoms (Table 4).

Parental allergic disease was a general risk factor for wheezing, eczema and both in all age groups. Paternal and maternal histories of asthma and atopic dermatitis were specific risk factors for wheezing and eczema respectively. In particular, the correlation was high between wheezing in infants and maternal history of asthma in all age groups, as well as between

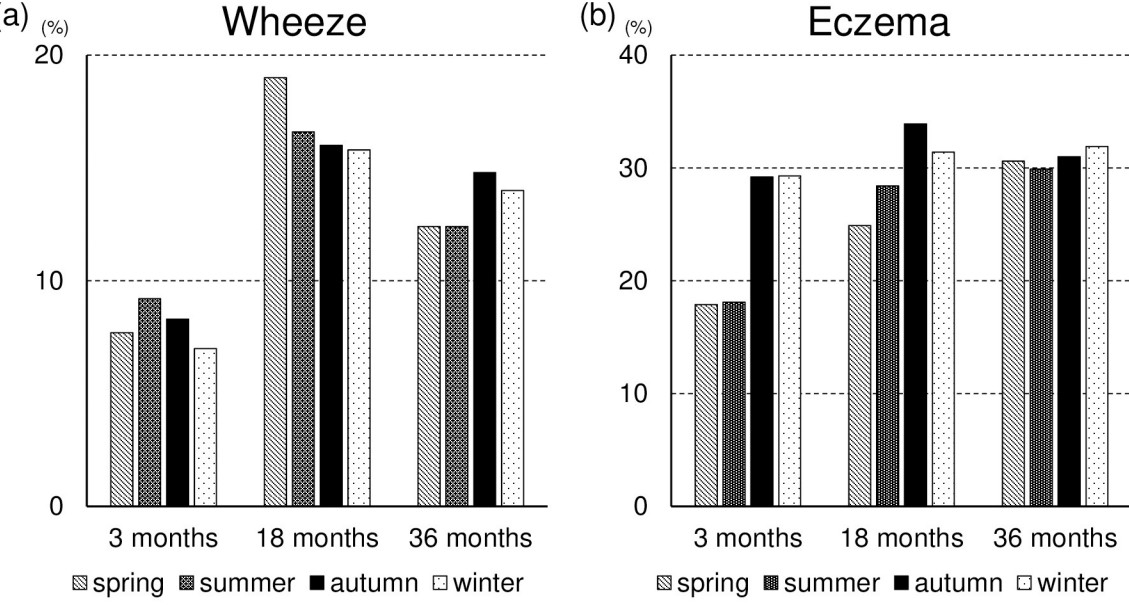

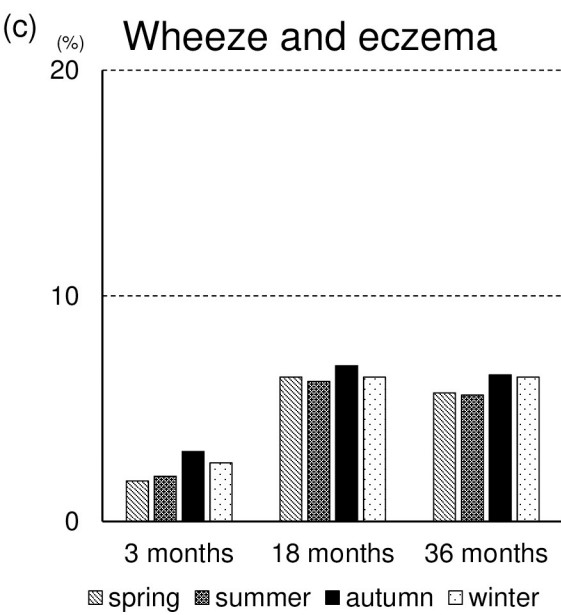

**Fig 2.** Prevalence of (a) wheezing, (b) eczema and (c) both wheezing and eczema in each season. Spring (months of March, April, and May), summer (months of June, July, and August, autumn (months of September, October, and November), and winter (months of December, January, and February).

eczema and paternal history of atopic dermatitis in the younger age groups. Each correlation on wheezing with a parental allergy history was larger in the older group of infants than in the younger group of infants. In terms of eczema, the correlation only with maternal atopic dermatitis was larger in the older group.

The statistically significant differences found in the univariate analyses were maintained in multivariable analysis of the data (Tables 5–7).

**Table 2. Univariate analysis between wheeze and covariates in three age groups.**

| | | 3 month group | | | 18 month group | | | 36 month group | | |
|---|---|---|---|---|---|---|---|---|---|---|
| | | OR | 95%CI | P | OR | 95%CI | P | OR | 95%CI | P |
| Gender | | | | | | | | | | |
| | Female | reference | – | – | reference | – | – | reference | – | – |
| | Male | 1.51 | 1.40–1.63 | <0.001 | 1.43 | 1.36–1.51 | <0.001 | 1.37 | 1.29–1.46 | <0.001 |
| Birth season | | | | | | | | | | |
| | Spring (Mar. to May) | reference | – | – | reference | – | – | reference | – | – |
| | Summer (Jun. to Aug.) | 1.23 | 1.10–1.36 | <0.001 | 0.85 | 0.79–0.92 | <0.001 | 1.00 | 0.91–1.09 | 0.981 |
| | Autumn (Sep. to Nov.) | 1.09 | 0.98–1.21 | 0.118 | 0.81 | 0.75–0.88 | <0.001 | 1.22 | 1.12–1.33 | <0.001 |
| | Winter (Dec. to Feb.) | 0.91 | 0.81–1.01 | 0.084 | 0.80 | 0.74–0.87 | <0.001 | 1.14 | 1.05–1.25 | 0.003 |
| Birth order | | | | | | | | | | |
| | First | reference | – | – | reference | – | – | reference | – | – |
| | Second and after | 1.21 | 1.12–1.31 | <0.001 | 1.05 | 1.00–1.11 | 0.07 | 0.96 | 0.90–1.02 | 0.208 |
| Paternal bronchial asthma | | | | | | | | | | |
| | No | reference | – | – | reference | – | – | reference | – | – |
| | Yes | 1.45 | 1.29–1.64 | <0.001 | 1.75 | 1.60–1.92 | <0.001 | 2.40 | 2.18–2.64 | <0.001 |
| Maternal bronchial asthma | | | | | | | | | | |
| | No | reference | – | – | reference | – | – | reference | – | – |
| | Yes | 1.75 | 1.56–1.96 | <0.001 | 2.16 | 1.98–2.36 | <0.001 | 2.79 | 2.55–3.05 | <0.001 |
| Parental allergic disease | | | | | | | | | | |
| | No | reference | – | – | reference | – | – | reference | – | – |
| | Yes | 1.36 | 1.24–1.49 | <0.001 | 1.50 | 1.41–1.60 | <0.001 | 1.83 | 1.70–1.97 | <0.001 |

## Discussion

Our study showed the prevalence of infantile wheeze and eczema at 3, 18, and 36 months of age in a metropolitan city of Japan. We also identified the prevalence of allergic diseases in the working population and the different risk factors in each of the age groups.

Half of the population of Japan is expected to have allergic diseases [10]. A nationwide birth cohort study in Japan reported the prevalence of allergic diseases was 51% in mothers and 43% in fathers [11]. In the present study, one-quarter of the infants did not have a parental history of allergies, which is consistent with the expectation that approximately half of the mothers and half of the fathers had allergic diseases.

The original ISAAC survey found that asthma and atopic dermatitis are more common diseases in the younger age group of schoolchildren [12]. Piedimonte and Perez found that 1-year-old infants had more wheezing due to respiratory syncytial virus, rhinovirus, or other infections than school-aged children [13]. In the present survey, the prevalence of wheezing was lowest in the 3-month age group and highest in the 18-month age group.

The T-CHILD survey reported the incidence of infantile wheezing in a hospital-based birth cohort study in Tokyo, Japan [14]. It reported wheezing in 20% and 16% of the children who were aged 1 and 3 years, respectively, using modified ISAAC questionnaire data, which showed a higher incidence rate than the present study. The parents of the participants in the T-CHILD study also had a higher prevalence of asthma than those in the present study. In the present study, parental asthma was found to be a risk factor for wheezing in children. We speculate that the higher prevalence of wheezing in children in the T-CHILD study was due to the higher prevalence of asthma in that parent group.

We found that the prevalence of eczema was higher in older children than younger ones. In this study, we defined eczema as a recurrent itchy rash. Parental reported eczema has good

**Table 3. Univariate analysis between eczema and covariates in three age groups.**

| | | 3 month group | | | 18 month group | | | 36 month group | | |
|---|---|---|---|---|---|---|---|---|---|---|
| | | OR | 95%CI | P | OR | 95%CI | P | OR | 95%CI | P |
| Gender | | | | | | | | | | |
| | Female | reference | – | – | reference | – | – | reference | – | – |
| | Male | 1.23 | 1.17–1.29 | <0.001 | 1.09 | 1.05–1.14 | <0.001 | 1.37 | 1.29–1.46 | <0.001 |
| Birth season | | | | | | | | | | |
| | Spring (Mar. to May) | reference | – | – | reference | – | – | reference | – | – |
| | Summer (Jun. to Aug.) | 1.02 | 0.94–1.10 | 0.657 | 1.20 | 1.12–1.28 | <0.001 | 1.00 | 0.91–1.09 | 0.981 |
| | Autumn (Sep. to Nov.) | 1.90 | 1.77–2.04 | <0.001 | 1.54 | 1.45–1.65 | <0.001 | 1.22 | 1.12–1.33 | <0.001 |
| | Winter (Dec. to Feb.) | 1.91 | 1.78–2.04 | <0.001 | 1.38 | 1.29–1.47 | <0.001 | 1.14 | 1.05–1.25 | 0.003 |
| Birth order | | | | | | | | | | |
| | First | reference | – | – | reference | – | – | reference | – | – |
| | Second and after | 0.77 | 0.73–0.81 | <0.001 | 0.90 | 0.86–0.94 | <0.001 | 0.96 | 0.90–1.02 | 0.208 |
| Paternal atopic dermatitis | | | | | | | | | | |
| | No | reference | – | – | reference | – | – | reference | – | – |
| | Yes | 1.83 | 1.71–1.96 | <0.001 | 2.33 | 2.18–2.48 | <0.001 | 2.31 | 2.16–2.47 | <0.001 |
| Maternal atopic dermatitis | | | | | | | | | | |
| | No | reference | – | – | reference | – | – | reference | – | – |
| | Yes | 1.70 | 1.61–1.81 | <0.001 | 2.10 | 1.98–2.22 | <0.001 | 2.34 | 2.21–2.48 | <0.001 |
| Parental allergic disease | | | | | | | | | | |
| | No | reference | – | – | reference | – | – | reference | – | – |
| | Yes | 1.70 | 1.60–1.80 | <0.001 | 2.00 | 1.90–2.12 | <0.001 | 1.83 | 1.70–1.97 | <0.001 |

**Table 4. Univariate analysis between wheeze and eczema and covariates in three age groups.**

| | | 3 month group | | | 18 month group | | | 36 month group | | |
|---|---|---|---|---|---|---|---|---|---|---|
| | | OR | 95%CI | P | OR | 95%CI | P | OR | 95%CI | P |
| Gender | | | | | | | | | | |
| | Female | reference | – | – | reference | – | – | reference | – | – |
| | Male | 1.62 | 1.41–1.86 | <0.001 | 1.42 | 1.31–1.55 | <0.001 | 1.35 | 1.23–1.47 | <0.001 |
| Birth season | | | | | | | | | | |
| | Spring (Mar. to May) | reference | – | – | reference | – | – | reference | – | – |
| | Summer (Jun. to Aug.) | 1.11 | 0.90–1.37 | 0.334 | 0.98 | 0.87–1.10 | 0.703 | 0.99 | 0.87–1.12 | 0.851 |
| | Autumn (Sep. to Nov.) | 1.74 | 1.43–2.12 | <0.001 | 1.08 | 0.97–1.22 | 0.173 | 1.15 | 1.02–1.30 | 0.024 |
| | Winter (Dec. to Feb.) | 1.45 | 1.19–1.78 | <0.001 | 1.00 | 0.89–1.13 | 0.961 | 1.12 | 0.99–1.28 | 0.068 |
| Birth order | | | | | | | | | | |
| | First | reference | – | – | reference | – | – | reference | – | – |
| | Second and after | 1.07 | 0.93–1.22 | 0.356 | 1.05 | 0.96–1.14 | 0.292 | 1.00 | 0.92–1.09 | 0.946 |
| Paternal allergic disease | | | | | | | | | | |
| | No | reference | – | – | reference | – | – | reference | – | – |
| | Yes | 1.82 | 1.52–2.18 | <0.001 | 2.08 | 1.86–2.32 | <0.001 | 2.56 | 2.27–2.89 | <0.001 |

**Table 5. Multivariable analysis between wheeze and covariates in the three age groups.**

| | | 3 month group | | | 18 month group | | | 36 month group | | |
|---|---|---|---|---|---|---|---|---|---|---|
| | | aOR | 95%CI | P | aOR | 95%CI | P | aOR | 95%CI | P |
| Gender | | | | | | | | | | |
| | Female | reference | – | – | reference | – | – | reference | – | – |
| | Male | 1.51 | 1.40–1.63 | <0.001 | 1.43 | 1.36–1.52 | <0.001 | 1.37 | 1.29–1.46 | <0.001 |
| Birth season | | | | | | | | | | |
| | Spring (Mar. to May) | reference | – | – | reference | – | – | reference | – | – |
| | Summer (Jun. to Aug.) | 1.22 | 1.10–1.36 | <0.001 | 0.85 | 0.79–0.92 | <0.001 | 1.00 | 0.92–1.10 | 0.945 |
| | Autumn (Sep. to Nov.) | 1.09 | 0.98–1.22 | 0.103 | 0.82 | 0.76–0.88 | <0.001 | 1.23 | 1.12–1.34 | <0.001 |
| | Winter (Dec. to Feb.) | 0.91 | 0.81–1.02 | 0.093 | 0.81 | 0.75–0.87 | <0.001 | 1.14 | 1.04–1.25 | 0.004 |
| Birth order | | | | | | | | | | |
| | First | reference | – | – | reference | – | – | reference | – | – |
| | Second and after | 1.22 | 1.13–1.32 | <0.001 | 1.06 | 1.00–1.12 | 0.042 | 0.98 | 0.92–1.04 | 0.427 |
| Paternal bronchial asthma | | | | | | | | | | |
| | No | reference | – | – | reference | – | – | reference | – | – |
| | Yes | 1.41 | 1.25–1.60 | <0.001 | 1.71 | 1.56–1.87 | <0.001 | 2.33 | 2.11–2.56 | <0.001 |
| Maternal bronchial asthma | | | | | | | | | | |
| | No | reference | – | – | reference | – | – | reference | – | – |
| | Yes | 1.74 | 1.55–1.95 | <0.001 | 2.12 | 1.94–2.32 | <0.001 | 2.75 | 2.51–3.01 | <0.001 |

diagnostic precision [15]; however, it is noted that caregivers cannot always accurately assess infants' medical conditions [16]. We believe that some parents may have underestimated itching in younger children.

The prevalence of wheezing differed by both season and age. This is most likely the effect of various respiratory infections. In particular, in the 18-month age group, a higher prevalence of wheeze was observed in children born in spring than in those born in other seasons. The recall

**Table 6. Multivariable analysis between eczema and covariates in the three age groups.**

| | | 3 month group | | | 18 month group | | | 36 month group | | |
|---|---|---|---|---|---|---|---|---|---|---|
| | | aOR | 95%CI | P | aOR | 95%CI | P | aOR | 95%CI | P |
| Gender | | | | | | | | | | |
| | Female | reference | – | – | reference | – | – | reference | – | – |
| | Male | 1.24 | 1.18–1.30 | <0.001 | 1.10 | 1.05–1.15 | <0.001 | 1.04 | 1.00–1.09 | 0.080 |
| Birth season | | | | | | | | | | |
| | Spring (Mar. to May) | reference | – | – | reference | – | – | reference | – | – |
| | Summer (Jun. to Aug.) | 1.01 | 0.94–1.09 | 0.719 | 1.21 | 1.13–1.29 | <0.001 | 0.96 | 0.90–1.03 | 0.269 |
| | Autumn (Sep. to Nov.) | 1.92 | 1.79–2.06 | <0.001 | 1.56 | 1.47–1.67 | <0.001 | 1.02 | 0.95–1.09 | 0.568 |
| | Winter (Dec. to Feb.) | 1.93 | 1.80–2.08 | <0.001 | 1.40 | 1.31–1.49 | <0.001 | 1.07 | 1.00–1.14 | 0.050 |
| Birth order | | | | | | | | | | |
| | First | reference | – | – | reference | – | – | reference | – | – |
| | Second and after | 0.78 | 0.74–0.82 | <0.001 | 0.92 | 0.88–0.96 | <0.001 | 0.95 | 0.91–1.00 | 0.034 |
| Paternal atopic dermatitis | | | | | | | | | | |
| | No | reference | – | – | reference | – | – | reference | – | – |
| | Yes | 1.78 | 1.66–1.90 | <0.001 | 2.26 | 2.12–2.41 | <0.001 | 2.23 | 2.08–2.39 | <0.001 |
| Maternal atopic dermatitis | | | | | | | | | | |
| | No | reference | – | – | reference | – | – | reference | – | – |
| | Yes | 1.66 | 1.57–1.77 | <0.001 | 2.05 | 1.93–2.17 | <0.001 | 2.28 | 2.15–2.41 | <0.001 |

**Table 7. Multivariable analysis between wheeze and eczema and covariates in the three age groups.**

| | | 3 month group | | | 18 month group | | | 36 month group | | |
|---|---|---|---|---|---|---|---|---|---|---|
| | | aOR | 95%CI | P | aOR | 95%CI | P | aOR | 95%CI | P |
| Gender | | | | | | | | | | |
| | Female | reference | – | – | reference | – | – | reference | – | – |
| | Male | 1.62 | 1.41–1.87 | <0.001 | 1.42 | 1.30–1.54 | <0.001 | 1.35 | 1.24–1.48 | <0.001 |
| Birth season | | | | | | | | | | |
| | Spring (Mar. to May) | reference | – | – | reference | – | – | reference | – | – |
| | Summer (Jun. to Aug.) | 1.10 | 0.89–1.37 | 0.364 | 0.98 | 0.87–1.10 | 0.745 | 1.00 | 0.88–1.14 | 0.954 |
| | Autumn (Sep. to Nov.) | 1.74 | 1.43–2.11 | <0.001 | 1.08 | 0.96–1.21 | 0.219 | 1.17 | 1.03–1.32 | 0.013 |
| | Winter (Dec. to Feb.) | 1.45 | 1.18–1.77 | <0.001 | 1.00 | 0.89–1.13 | 0.962 | 1.14 | 1.00–1.29 | 0.043 |
| Birth order | | | | | | | | | | |
| | First | reference | – | – | reference | – | – | reference | – | – |
| | Second and after | 1.08 | 0.94–1.24 | 0.271 | 1.06 | 0.97–1.15 | 0.190 | 1.01 | 0.93–1.11 | 0.746 |
| Paternal allergic disease | | | | | | | | | | |
| | No | reference | – | – | reference | – | – | reference | – | – |
| | Yes | 1.81 | 1.51–2.17 | <0.001 | 2.07 | 1.86–2.32 | <0.001 | 2.56 | 2.27–2.90 | <0.001 |

bias of parents might have affected their answers regarding wheeze-inducing infections in a previous winter. The difference in wheezing at 36 months of age seems to indicate seasonal differences in pediatric asthma in Japan [17].

We found eczema was more common in children born in autumn and winter than those born in spring or summer. In particular, the pattern of higher rates in these seasons was striking in the 3-month age group, which was consistent with a review that Calov *et al.* reported where they found that children born in autumn and winter in the northern hemisphere have an increased risk of atopic dermatitis [18].

In the first 3 months of life, the effects of both climate and genetic predisposition are considered to be stronger than those of ultraviolet rays or nutrition [19]. In newborn babies up to 3 months of age, eczema is likely to be exacerbated by their immature skin barrier function. Infants with early-onset eczema have been shown to have an increased risk of subsequent food allergies [20]. Our results that infants born in autumn or winter had higher prevalence of eczema in the 3-month-old group support the findings of another study that the prevalence of food allergies was higher in infants born in autumn or winter [21].

A parental history of atopic diseases is a known risk factor for the onset of atopic diseases in their children [22, 23]. In the present study, the prevalence of wheezing and eczema in the children was significantly associated with a parental history of each atopic disease, and the correlation on wheezing was stronger in older children. Yang *et al.* described a number of phenotype classifications of wheezing and reported that transient early wheezing was associated with a lower incidence of parental history of asthma than the other phenotypes and persistent wheezing was associated with a higher incidence of parental history of asthma [14]. Due to the high correlation of wheezing with parental history of asthma in the older children in the present study, the data might include more transient early wheezing in the 18-month group and more persistent wheezing in the 36-month group. In the present survey, the prevalence of transient wheezing might appear to have peaked between the ages of 3 and 18 months.

Yamamoto-Hanada *et al.* found that the correlation of parental history of atopic dermatitis in the persistent phenotype of eczema was higher than that of other phenotypes [24]. In the present study, the correlation of parental atopic dermatitis was high in the 18-month and 36-month age group, so it appears that the onset of the persistent phenotype increased in

babies born in spring or summer and contributed to the increase in eczema prevalence in these age groups.

## Strengths and limitations

This survey had a high response rate and covered almost the entire population of the target age groups in a large city. Previous studies from various regions of Japan have also reported the prevalence of allergies in infants; however, most of them were conducted in a confined cohort, such as hospitalized patients or children in a nursery [25–27]. Other surveys were conducted in the general population but included only randomly selected children [28]. We believe that the results of this study, therefore, provide a much higher level of confidence regarding the prevalence of allergies in infants in Japan compared to previous studies.

This study has some limitations. This was a cross-sectional survey of three age groups. We could not confirm the causal relationship between allergic symptoms and risk factors or predict the infantile trajectories of wheezing and eczema.

We used the modified ISAAC questionnaire to assess parent-reported allergy symptoms instead of using a doctor's diagnosis. The ISAAC questionnaire has been validated only for children aged 6–7 and 13–14 years. We did not have access to any validation study of questionnaires for infants. However, we used the same modified ISAAC scale for younger children that was also used in other studies conducted in Japan[14, 24, 28].

In the future, we plan to conduct a longitudinal survey to investigate the causal relationships involved in the development of allergies from various factors.

## Conclusion

We identified the prevalence of infantile wheezing and eczema in three age groups of infants through a parental census survey. The prevalence of both symptoms in males was significantly higher than that in females. Participants born in autumn and winter had a higher incidence of eczema in each age group. The data also showed a strong association of wheezing with parental allergic diseases in children in the older age group.

## Acknowledgments

We are grateful to all participants and parents attending the group health checkups in Nagoya who responded to the survey questionnaire. We also thank the staff of the Division of Environment Disaster and Health, Environmental Bureau of Nagoya, Japan.

## Author Contributions

**Conceptualization:** Masaki Futamura, Komei Ito.

**Data curation:** Masaki Futamura, Yoshimichi Hiramitsu, Naomi Kamioka, Chikae Yamaguchi, Harue Umemura, Shiro Sugiura, Yasuto Kondo, Komei Ito.

**Formal analysis:** Yoshimichi Hiramitsu.

**Investigation:** Masaki Futamura.

**Methodology:** Masaki Futamura, Yoshimichi Hiramitsu.

**Validation:** Yoshimichi Hiramitsu.

**Writing – original draft:** Masaki Futamura.

**Writing – review & editing:** Masaki Futamura, Yoshimichi Hiramitsu, Naomi Kamioka, Chikae Yamaguchi, Harue Umemura, Rieko Nakanishi, Shiro Sugiura, Yasuto Kondo, Komei Ito.

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
