## [Decision Letter · Decision Letter 0]

24 Feb 2022

PONE-D-22-01155Prevalence of infantile wheezing and eczema in a metropolitan city in Japan: a complete census surveyPLOS ONE

Dear Dr. Futamura,

Thank you for submitting your manuscript to PLOS ONE. After careful consideration, we feel that it has merit but does not fully meet PLOS ONE’s publication criteria as it currently stands. Therefore, we invite you to submit a revised version of the manuscript that addresses the points raised during the review process.

We look forward to receiving your revised manuscript.

Kind regards,

Linglin Xie

Academic Editor

PLOS ONE

Journal Requirements:

Reviewers' comments:

Reviewer's Responses to Questions

**Comments to the Author**

1. Is the manuscript technically sound, and do the data support the conclusions?

Reviewer #1: Yes

2. Has the statistical analysis been performed appropriately and rigorously? 

Reviewer #1: Yes

3. Have the authors made all data underlying the findings in their manuscript fully available?

Reviewer #1: No

4. Is the manuscript presented in an intelligible fashion and written in standard English?

Reviewer #1: Yes

5. Review Comments to the Author

Reviewer #1: Short Title: Infantile wheezing and eczema in Japan

In this manuscript, Futamura et al. presented a good study based on large-scale survey data that could serve as a basis for generalization. The results are large population-based, therefore providing enough confidence level for predicting the infantile allergic symptoms (wheeze and eczema) in Nagoya, Japan. The statistical methods are proper, and the results are pretty solid. There are some parts that still need to be clarified, listed in the comment section point-by-point.

Major

--The prevalence of both eczema and wheeze is well put in Table 1. It might be interesting to analyze the group with both eczema and wheeze symptoms (if the data are available from the questionnaire). Infants with both symptoms would have a higher probability of developing asthma than those groups that show eczema alone or wheeze alone. Would this combined prevalence be associated with age, sex/gender, birth order, birth season, and parental allergy history?

--In addition to my first comment, I would like to see the prevalence of the combined group since Figure 2 shows some interesting results and differences between the two symptoms, depending on the birth season as well as among the three age groups.

--On Page 14, “Each correlation with parental allergy history was larger in the older group of infants than in the younger groups.” While a monotonically increasing trend is generally observed in Table 2 (Wheeze, 1.45-1.75-2.40, 1.75-2.16-2.79, 1.36-1.50-1.83), Table 3 does not necessarily show this trend across all age groups (1.83-2.33-2.31, 1.70-2.10-2.34, 1.70-2.00-1.83). For instance, comparing 18-month group and 36-month group, it shows a slight decrease in the correlation between eczema and parental allergic disease. Please justify this. Also note that this does not affect the general conclusion of “a strong association with parental allergic diseases in children in the older age group”.

--If the ISAAC survey data is only validated for age 6-7 and 13-14 years, how did the authors scale or normalize the age for younger children? In the last section, the explanation is vague and seems unclear to me.

--In your conclusion, gender seems to be missing. But Tables 2-5 showed significant differences between male and female participants.

Minor

--et al should be “et al.” This should be corrected throughout the manuscript (e.g., Pages 18-20).

6. PLOS authors have the option to publish the peer review history of their article (what does this mean?). If published, this will include your full peer review and any attached files.

Reviewer #1: No

---

## [Author Response · Author response to Decision Letter 0]

16 Mar 2022

Reviewer #1: Major comments:

The prevalence of both eczema and wheeze is well put in Table 1. It might be interesting to analyze the group with both eczema and wheeze symptoms (if the data are available from the questionnaire). Infants with both symptoms would have a higher probability of developing asthma than those groups that show eczema alone or wheeze alone. Would this combined prevalence be associated with age, sex/gender, birth order, birth season, and parental allergy history?

RESPONSE: We appreciate your suggestion. We have analyzed the data and added and amended the sentences as follows:

(Line 152) “In infants, the prevalence rates of wheezing were 8%, 17%, and 13%, and those of eczema were 24%, 30%, and 31%, and those of both symptoms were 2%, 7%, and 6% at 3, 18, and 36 months, respectively.”

(Line 166) “The tendency in eczema was also observed in both symptoms.”

(Line 184) “The significant difference in the first-born children disappeared for both symptoms (Table 4).”

(Line 195) “Parental allergic disease was a general risk factor for both wheezing, and eczema, and both in all age groups.”

However, we cannot predict the future asthma prevalence in infants with wheezing and eczema because this is a cross-sectional population survey. 

According to the analysis, we have shown the prevalence of both wheezing and eczema in revised Table 1 and added two tables – new Tables 4 and 7.

--In addition to my first comment, I would like to see the prevalence of the combined group since Figure 2 shows some interesting results and differences between the two symptoms, depending on the birth season as well as among the three age groups.

RESPONSE: We have shown the difference in the prevalence of birth seasons in Figure 2. 

--On Page 14, “Each correlation with parental allergy history was larger in the older group of infants than in the younger groups.” While a monotonically increasing trend is generally observed in Table 2 (Wheeze, 1.45-1.75-2.40, 1.75-2.16-2.79, 1.36-1.50-1.83), Table 3 does not necessarily show this trend across all age groups (1.83-2.33-2.31, 1.70-2.10-2.34, 1.70-2.00-1.83). For instance, comparing 18-month group and 36-month group, it shows a slight decrease in the correlation between eczema and parental allergic disease. Please justify this. Also note that this does not affect the general conclusion of “a strong association with parental allergic diseases in children in the older age group”.

RESPONSE: We agree that odds ratios of eczema in the 36-month group were not higher than those in the 18-month group. We have amended the sentences as follows:

(Line 200) “Each correlation on wheezing with a parental allergy history was larger in the older group of infants than in the younger group of infants. In terms of eczema, the correlation only with maternal atopic dermatitis was larger in the older group.”

(Line 280) “and the correlation on wheezing was stronger in older children.”

(Line 325) “The data also showed a strong association of wheezing with parental allergic diseases in children in the older age group.”

--If the ISAAC survey data is only validated for age 6-7 and 13-14 years, how did the authors scale or normalize the age for younger children? In the last section, the explanation is vague and seems unclear to me.

RESPONSE: We did not validate the modified questionnaire for infants. However, it was already used in other surveys. We have amended the sentences in the Discussion as follows:

(Line 233) “The original ISAAC survey revealed that asthma and atopic dermatitis are more common diseases in the younger age group of schoolchildren”

(Line 241) “respectively, using modified ISAAC questionnaire data”

(Line 313) “We did not have access to any validation study of validated questionnaires for infants. Therefore However, we used the same modified the ISAAC scale for younger children that was also used in as other studies conducted in Japan have also done.”

--In your conclusion, gender seems to be missing. But Tables 2-5 showed significant differences between male and female participants.

RESPONSE: We appreciate you for pointing out the gender difference. We have changed the sentence in the conclusion as follows:

(Line 323) “The prevalence of both symptoms in males was significantly higher than that in females.”

Minor

--et al should be “et al.” This should be corrected throughout the manuscript (e.g., Pages 18-20).

RESPONSE: Thank you for notifying this issue. We have corrected the grammatical expression in the text.

In addition, we found the word was used incorrectly. Therefore, we have now revised it – from “multivariate” to “multivariable.”

Ref. Hidalgo B, Goodman M. Multivariate or multivariable regression? Am J Public Health. 2013 Jan;103(1):39-40. PMID: 23153131

---

## [Decision Letter · Decision Letter 1]

22 Apr 2022

Prevalence of infantile wheezing and eczema in a metropolitan city in Japan: a complete census survey

PONE-D-22-01155R1

Dear Dr. Futamura,

We’re pleased to inform you that your manuscript has been judged scientifically suitable for publication and will be formally accepted for publication once it meets all outstanding technical requirements.

Kind regards,

Linglin Xie

Academic Editor

PLOS ONE

Additional Editor Comments (optional):

Reviewers' comments:

Reviewer's Responses to Questions

**Comments to the Author**

1. If the authors have adequately addressed your comments raised in a previous round of review and you feel that this manuscript is now acceptable for publication, you may indicate that here to bypass the “Comments to the Author” section, enter your conflict of interest statement in the “Confidential to Editor” section, and submit your "Accept" recommendation.

Reviewer #1: All comments have been addressed

2. Is the manuscript technically sound, and do the data support the conclusions?

Reviewer #1: Yes

3. Has the statistical analysis been performed appropriately and rigorously? 

Reviewer #1: Yes

4. Have the authors made all data underlying the findings in their manuscript fully available?

Reviewer #1: No

5. Is the manuscript presented in an intelligible fashion and written in standard English?

Reviewer #1: Yes

6. Review Comments to the Author

Reviewer #1: All my comments are properly addressed.

The manuscript is technically sound, and analyses are well-put.

7. PLOS authors have the option to publish the peer review history of their article (what does this mean?). If published, this will include your full peer review and any attached files.

Reviewer #1: No